# Study on the Synthesis of High-Purity γ-Phase Mesoporous Alumina with Excellent CO_2_ Adsorption Performance via a Simple Method Using Industrial Aluminum Oxide as Raw Material

**DOI:** 10.3390/ma14195465

**Published:** 2021-09-22

**Authors:** Zhonglin Li, Ding Wang, Jialong Shen, Junxue Chen, Chengzhi Wu, Zizheng Qu, Kun Luo, Zhengbing Meng, Yibing Li

**Affiliations:** 1Department of Materials Science and Engineering, Guilin University of Technology, Guilin 541000, China; dahe121133@gmail.com (Z.L.); dingnvhuang@gmail.com (D.W.); Jialong.Shen@glut.edu.cn (J.S.); cchenjunxue@gmail.com (J.C.); wuchengzhi73@gmail.com (C.W.); dadpro01@gmail.com (Z.Q.); 2Collaborative Innovation Center for Exploration of Nonferrous Metal Deposits and Efficient Utilization of Resources, Guilin University of Technology, Guilin 541000, China; 3Key Laboratory of New Processing Technology for Nonferrous Metals and Materials, Ministry of Education, Guilin University of Technology, Guilin 541000, China; 4School of Materials Science and Engineering, Changzhou University, Changzhou 213614, China

**Keywords:** CO_2_ adsorption, mesoporous *γ*-alumina, high purity, direct aging method, ammonium salt substitution method

## Abstract

To mitigate the global greenhouse effect and the waste of carbon dioxide, a chemical raw material, high-purity γ-phase mesoporous alumina (MA) with excellent CO_2_ adsorption performance was synthesized by the direct aging method and ammonium salt substitution method. With this process, not only can energy consumption and time be shortened to a large extent but the final waste can also be recycled to the mother liquor by adding calcium hydroxide. Reaction conditions, i.e., pH value, calcination temperature, and desodium agent, were investigated in detail with the aid of X-ray fluorescence spectrum (XRF), X-ray diffraction (XRD), scanning electron microscopy (SEM), Brunauer–Emmett–Teller (BET) and Barret-Joyner-Hallender (BJH) methods, nonlocal density functional theory (NLDFT), transmission electron microscopy (TEM), temperature-programmed desorption of CO_2_ (CO_2_-TPD), and presented CO_2_ adsorption measurement. The results of this study are summarized as follows: the impurity content of the MA synthesized under optimal conditions is less than 0.01%, and its total removal rate of impurities is 99.299%. It was found that the MA adsorbent has a large specific surface area of 377.8 m^2^/g, pore volume of 0.55 cm^3^/g, and its average pore diameter is 3.1 nm. Under the condition of a gas flow rate of 20 cm^3^/min, its CO_2_ adsorption capacity is 1.58 mmol/g, and after 8 times of cyclic adsorption, the amount of CO_2_ adsorption remained basically unchanged, both of which indicate that the material has excellent adsorption properties and can be widely used for the adsorption of carbon dioxide.

## 1. Introduction

Greenhouse gas emissions have been a worldwide problem due to industrialization for the past decades. CO_2_ contributes the most to the greenhouse effect, while it is one of the important raw materials in industrial fields such as metallurgy, chemical industry, building materials, and medical treatment. It is important that CO_2_ can be effectively captured and converted to useful organic matter [1,2,3,4]. At present, CO_2_ capture methods mainly include membrane separation [5], absorption [6,7,8,9], and adsorption methods [7,9]. the use of solid adsorbents for chemical adsorption of CO_2_ has excellent adsorption capacity, faster adsorption rate, and higher selectivity.

The *γ*-phase mesoporous alumina (MA) is usually presented as a material with a large specific surface area and excellent chemical activity [10,11,12,13,14]. It has been used extensively in adsorbents, filters, catalysts, and catalyst support. *γ*-Al_2_O_3_ with larger specific surface area and higher pore volume is favorable to attract reactants flowing into catalytic and adsorb basic sites. It also facilitates the formation of active sites per unit area and therefore is beneficial to the catalytic activity and the adsorption property [15,16,17,18,19,20]. Chemical preparation methods of the *γ*-phase MA have been reported, such as sol–gel, precipitation, gas phase deposition, and hydrometallurgy [21,22,23,24,25,26,27,28,29,30,31,32]. MA was used to adsorb dye [33,34], Cr ions, heavy metals [35,36,37], and CO_2_ [35,38,39,40,41,42,43]. The capture of CO_2_ by using MA has been extensively studied. For example, Cai et al. [35] utilized P123 as a template to synthesize alumina and modified it with tetraethylenepentamine (TEPA). It showed that the MA has a CO_2_ adsorption capacity of 0.7 mmol/g at 25 °C, greatly increasing the industrial alumina adsorption capacity (from 0.07 to 0.7 mmol/g). Sakwanovak et al. [38] synthesized monolithic alumina honeycombs using polyethyleneimine (PEI) as an amino reagent. The monolithic adsorbent maintains 0.7 mmol/g in the CO_2_ adsorption cycle. Gunathilake et al. [40] used boehmite (Bh) and (3-cyanopropyl) triethoxysilane (CPTS: 13 mmol) as precursors to prepare an alumina adsorbent (Bh-AO13), which was endowed with amidoxime groups (AO). It found that the CO_2_ adsorption capacity of Bh-AO13 at 25 °C and 120 °C were 0.45 mmol/g and 3.84 mmol/g, respectively, which is better than the result of MA-Bh (the CO_2_ adsorption were 0.58 mmol/g and 2.17 mmol/g at a temperature of 25 °C and 120 °C, respectively). Bali et al. [41] used poly(ethyleneimine) (PEI) and poly(allylamine) (PAA) to modify alumina separately. At 110 °C, the CO_2_ adsorption capacity of the two modified aluminas was up to 1.87 mmol/g and 1.07 mmol/g, respectively.

Synthetic methods of excellent performance MA have been proposed, but no industrialized application has been reported, because of the high-cost raw materials, complex synthesis process, and that the final waste cannot be recycled. The present study focuses on synthesizing MA via novel and practical processes with lower-cost primary materials; the final waste can be recycled to the mother liquor by adding calcium hydroxide. The effects of certain crucial preparation parameters, such as pH value, calcination temperature, and desodium agent have been investigated. The crystalline structure, porosity, morphology, and adsorption performance of the as-prepared MA were characterized and presented.

## 2. Experimental Procedures

### 2.1. Materials

Industrial Al(OH)_3_ provided by an alumina company in Guangxi, China, was directly used without any pretreatment. The primary contents are listed in Table 1. The other reagents were purchased from Xi-long Chemical Co., Ltd., Shantou, China, without any pretreatment.

### 2.2. Preparation of Mesoporous Alumina Materials

A total of 219.4 g of NaOH solid was dissolved in 500 mL of high-purity water, and then 281.4 g of industrial Al(OH)_3_ was poured into the NaOH solution and stirred until all solids were dissolved under hydrothermal conditions of 95 °C. HCl was slowly added to the sodium aluminate solution in a water bath at 30 °C, accompanied by magnetic stirring. It was aged at 65 °C under vigorous stirring for 2 h at the required pH value (pH = 6, 8, 10, 12, and 13). The obtained colloidal substance was centrifuged and washed with high-purity water until the filtrate became neutral, then dispersed in the solid desodium agent (ammonium carbonate aqueous solution). After vigorous stirring for 4 h, a highly uniform mixture was formed, which was dried in an electric constant temperature air drying oven at 120 °C for 12 h. Finally, the calcination was carried out at 500 °C, 600 °C, 700 °C, 800 °C, 900 °C, and 1000 °C at a heating rate of 2 °C/min, and the holding time was 4 h.

### 2.3. Characterization of Alumina Materials

The impurity content of alumina (iron, alumina, and silicon) was measured by X-ray fluorescence spectrum analysis (XRF, XRF-1800, Shimadzu Japan Ltd., Shanghai, China). The morphology of the material was studied using a scanning electron microscope (SEM, S-4800, Hitachi Works, Ltd., Tokyo, Japan) worked at 5.0 KV and transmission electron microscopy (TEM, JEM-2100F, JEOL, Beijing, China). Structural phase analysis was carried out by X-ray diffraction (XRD) using Cu*Kα* radiation. The equipment was X’Pert PRO (PANalytical B.V., Almelo, The Netherlands), and the continuous mode was used to collect 2*θ* data from 10° to 80° with an 8°/min sampling pitch. Surface area and pore porosimetry analyzer NoVA 1200e (Quantachrome Instruments, Shanghai, China) was used to characterize nitrogen adsorption-desorption isotherms curve of all calcined MA materials at various partial pressures. Before BET and BJH/NLDFT measurements, all samples were degassed for 5 h at 200 °C. Using a surface-sensitive technique, temperature-programmed desorption of CO_2_ (CO_2_-TPD) was performed by AutoChem1 II 2920 (Micromeritics instrument Corp., Beijing, China) to probe the basic sites of the prepared material.

### 2.4. CO_2_ Adsorption Measurement

The method of CO_2_ sorption measurements was decided on the basis of reported literature [44,45,46,47,48] and applied with appropriate modifications. Approximately, 100 mg of solid adsorbent was placed in a clean and dry quartz tube centered in a heated oven on the chemisorber. Pure Argon gas was introduced at a gas flow rate of 80 cm^3^/min. The temperature was increased to 100 °C at a heating rate of 5 °C/min and was maintained for 60 min to remove impurities from the material surface. The temperature then was decreased to 75 °C, and the Argon gas was replaced with a mixture of CO_2_. Carbon dioxide adsorption was determined using a mixture of helium and carbon dioxide with a volume ratio of 5:95 at a specific temperature and atmospheric pressure (gas flow rate of 20 cm^3^/min and adsorption time of 30 min). During the adsorption process, the CO_2_ adsorption penetration curve of the adsorbent was obtained by measuring the CO_2_ concentration in the tail gas. The penetration curves were integrated to obtain the CO_2_ adsorption capacity of the solid adsorbent. After the CO_2_ adsorption was finished, the gas was replaced with pure Argon and the temperature was increased to 100 °C at a heating rate of 5 °C/min. It was kept at this temperature for 60 min for CO_2_ desorption. The same procedure was followed for cyclic adsorption. The desorption process (number of cycles: 8) was performed to determine the cyclic CO_2_ adsorption performance of the solid adsorbent.

## 3. Results and Discussion

### 3.1. Effects of Various Conditions on MA

#### 3.1.1. pH Value of the Solution

XRD patterns of mesoporous *γ*-Al_2_O_3_ particles synthesized in different pH solutions are shown in Figure 1A. It is shown that when pH value is lower than 12 (pH values of 6, 8, 10, 12), main diffraction peaks (111), (220), (311), (222), (400), (511) and (440) of *γ*-Al_2_O_3_ with a cubic structure [49] are indicated (JCPDS Card No. 10-0425), and there are no impurity diffraction peaks. This is evidence of high-purity *γ*-Al_2_O_3_ powders formation. When the solution pH value is 13, it shows the peak of *κ*-Al_2_O_3_, another phase of Al_2_O_3_.

Nitrogen adsorption–desorption isotherms and pore-size distribution curves of MA powder prepared in different pH solutions are shown in Figure 1B,C, respectively. According to the IUPAC (International Union of Pure and Applied Chemistry) classification [50], the nitrogen adsorption–desorption isotherms of powder materials prepared in pH solutions are IV isotherms, indicating all materials are MA. The adsorption isotherm has a slender H1 hysteresis loop, which is caused by the coverage of the adsorbate on the mesoporous pore wall. When the P/Po value is small, the amount of N_2_ adsorption is gradually increased with the increase in partial pressure and the presence of single-molecule adsorption of N_2_ on the pore surface; with the increase in P/Po, the capillary condensation phenomenon of N_2_ occurs in the pores. The surface adsorption starts from the monolayer to multiple layers with a sudden jump, and then the adsorption capacity increases rapidly. A subsequent long adsorption platform indicates that the adsorption of N_2_ in the capillary is saturated. When the pH of the solution exceeds 10, the adsorption is caused by the agglomeration of N_2_ between large particles. The measured specific surface area of the sample is relatively small, due to the small content of mesoporous channels in the sample. After reacting in different pH solutions, the physical properties of the alumina materials prepared by calcining at 500 °C are further characterized, as shown in Table 2. It can be seen when the pH value is lower than 10, the specific surface area of the MA increases from 263.4 m^2^/g to 377.8 m^2^/g, and the pore volume increases from 0.34 cm^3^/g to 0.55 cm^3^/g (pH value from 6 to 10). When the pH value exceeds 10, the specific surface area decreases from 377.8 m^2^/g to 221.8 m^2^/g, and the pore volume decreases from 0.55 cm^3^/g to 0.25 cm^3^/g (pH value from 10 to 13); this is because the higher the pH of the solution is, the more hydroxide ions will combine with the ammonium ions electrolyzed from the later-added solution of the sodium remover to form ammonia gas; therefore, the number of residual ammonium roots on the surface of MA material will be reduced. After roasting at a high temperature, the material surface has less pore structure, and the pore volume and specific surface area will decrease.

The CO_2_ adsorption performance of MA prepared under different pH conditions are shown in Table 3. It can be seen from the table that when the gas flow rate is constant, the CO_2_ adsorption capacity is proportional to the specific surface area of MA. CO_2_ can be captured at the surface of solid materials by both physical and chemical adsorption. Physical adsorption is mainly achieved by the van der Waals force between CO_2_ and the adsorbent surface. Therefore, the amount of CO_2_ adsorption is small due to the weak force between the adsorbent and the adsorbate. Chemical adsorption is mainly achieved by electron transfer, exchange, or sharing between CO_2_ and adsorbent surface atoms or molecules. The heat of adsorption is equivalent to the heat of the chemical reaction, and the selectivity is high. The CO_2_ adsorption capacity of the prepared MA is mainly dominated by chemical adsorption. The large specific surface area can give more basic sites on the surface of the MA material. All results show that the pH value of 10 is the optimum that can facilitate the synthesizing *γ*-Al_2_O_3_ powders which has a large specific surface area, a large pore volume, and excellent CO_2_ adsorption property.

#### 3.1.2. Calcination Temperature

XRD patterns of mesoporous *γ*-Al_2_O_3_ particles calcined at various temperatures are shown in Figure 2A. When the temperature ranges from 500 °C to 800 °C, diffraction peaks (111), (220), (311), (222), (400), (511) and (440) of *γ*-Al_2_O_3_ (JCPDS Card No. 10-0425) are shown, without any other diffraction peaks, which proves the existence of pure *γ*-Al_2_O_3_. However, when the temperature exceeds 800 °C (900 °C and 1000 °C), diffraction peaks of the other phases of alumina—*α*-Al_2_O_3_ (JCPDS Card No. 88-0826) and *θ*-Al_2_O_3_ (JCPDS Card No. 86-1410)–appear.

Nitrogen adsorption–desorption isotherms and BJH pore-size distribution curves of the materials calcined at various temperatures are shown in Figure 2B,C. Catalytic properties of the materials obtained in pH = 10 solution with various temperatures are shown in Table 4. The product calcined at 500 °C has a high specific surface area of 377.8 m^2^/g, a large pore volume of 0.55 cm^3^/g, and a pore size of 3.1 nm. The specific surface area and the pore volume decrease to 94.8 m^2^/g and 0.11 cm^3^/g, respectively, when the temperature increases from 500 °C to 1000 °C. The amount of nitrogen adsorbed is also gradually decreased. TEM images for various calcined temperatures are shown in Figure 3. It can be seen from the morphology that calcination at 500 °C can produce MA (particle size of which is within 100 nm) with larger specific surface area. As the calcination temperature rises, the particle size, the number of pores, and the pore diameter of the alumina material will increase. This is because under high-temperature conditions, the pore structure of the *γ*-Al_2_O_3_ material completely collapses, resulting in a rapid decrease in specific surface area and total pore volume. A dense structure of *α*-Al_2_O_3_ and *θ*-Al_2_O_3_ is therefore forming. *α*-Al_2_O_3_ belongs to the trigonal crystal system and is the most stable phase in aluminum oxides. It has a high melting point, high hardness, good wear resistance, high mechanical strength, good electrical insulation, corrosion resistance, etc. It is used to make pure aluminum, an ideal raw material for series of ceramics, abrasives, abrasive tools, and refractory materials. *θ*-Al_2_O_3_ is obtained by calcination of Bayer body aluminum hydroxide at 900–1100 °C, at a certain temperature rising rate. *θ*-Al_2_O_3_ belongs to the monoclinic system, and its properties are between *γ*-Al_2_O_3_ and *α*-Al_2_O_3_, and often coexists with *γ*-Al_2_O_3_ and *α*-Al_2_O_3_.

From the data in Table 4 and Table 5, it can be seen that the specific surface area decreases with higher calcination temperature, and the number of amino acids attached to the surface of the material will also decrease, which is not conducive to the physical and chemical adsorption of the material. In summary, the optimum temperature for obtaining high purity MA with a large specific surface area, large pore volume, and excellent CO_2_ adsorption property is 500 °C.

#### 3.1.3. Desodium Agent

The results indicate that after being aged for 2 h, some sodium impurities still remain in the resultant colloidal powders. The mechanism of the sodium removal agent is that NH_4_^+^ in the weak-acid ammonium salt reacts as a sodium removal agent and can replace the sodium ions remaining in *γ*-AlO(OH); sodium can be removed by washing with high-purity water, and a large amount of ammonium ion will remain on the surface of *γ*-AlO(OH) simultaneously. During the drying process, a large amount of ammonia gas is released leading to the formation of pores on the surface of *γ*-AlO(OH). After calcination at a high temperature, the alumina produced a large specific surface area and successfully doped many amine groups on its surface. Applying three kinds of weak-acid ammonium salt for desodium agents can completely remove the sodium impurities and effectively increase the specific surface area and pore volume of *γ*-Al_2_O_3_. The basic sites on the surface of the MA material are also increased. All these are beneficial to the CO_2_ adsorption capacity.

XRD patterns of MA particles with the aids of three desodium agents are shown in Figure 4A, and all the XRD spectra show only peaks of mesoporous alumina, indicating that the type of desodium agents has no effect on the purity of *γ*-Al_2_O_3_. It can be seen from Figure 4B and Table 6 that using desodium agents I and II can prepare highly porous *γ*-Al_2_O_3_ with specific surface areas of 278.1 m^2^/g and 248.3 m^2^/g, and pore volumes of 0.44 cm^3^/g and 0.47 cm^3^/g, respectively. Using the ammonium carbonate as a sodium removal agent can increase the specific surface area and pore volume more than using desodium agents I and II, which is conducive to improving the adsorption and catalytic performance of materials.

The CO_2_ adsorption capacity of industrial alumina and MA prepared by different desodium agents are shown in Table 7. The results indicate that when NH_4_Cl, CH_3_COONH_4_, and ammonium carbonate solution are used as desodium agents, the specific surface areas are 278.1 m^2^/g, 248.3 m^2^/g, and 377.8 m^2^/g, respectively. The CO_2_ adsorption capacities are 0.78 mmol/g, 1.03 mmol/g, and 1.58 mmol/g, respectively. Comparing the specific surface area and CO_2_ adsorption of NH_4_Cl and CH_3_COONH_4_ as sodium removal agents, it can be seen that MA particles with the larger specific surface area have less CO_2_ adsorption capacity instead. This is because the specific surface area and the number of surface basic sites are two important factors that affect the adsorption performance of CO_2_—the larger specific surface area is beneficial to improve the physical adsorption of the material, and the more basic sites on the surface, the stronger the chemisorption of CO_2_ by the MA material; thus, the number of basic sites per unit area of MA materials prepared with different desodium agents plays a dominant role in the adsorption capacity of CO_2_. While removing sodium impurities, different weak-acid ammonium salts are doped on the surface of the material, resulting in a different number of basic sites on the surface of the prepared MA material. (NH4)_2_CO_3_ solution is the most effective sodium removal agent, which helps to synthesize high-purity MA with large specific surface area and enhances the basic sites on the surface of MA materials, thus improving the physical and chemical adsorption of CO_2_.

### 3.2. The Synthesis of High-Purity Mesoporous γ-Al_2_O_3_

High purity mesoporous *γ*-Al_2_O_3_ materials with high porosity were successfully prepared via the direct aging method and the ammonium salt substitution method. As the sodium hydroxide solution dissolved the industrial aluminum hydroxide, soluble impurity Na_2_SiO_3_ was generated by the reaction of the SiO_2_ with NaOH liquor and entered into the NaAl(OH)_4_ solution. Na_2_SiO_3_ and NaAl(OH)_4_ continued to react to form precipitates, and most of the SiO_2_ impurities were then removed. However, there was only a trace amount of soluble Na_2_Zn(OH)_4_ flowing into the product, and all iron-containing compounds remain in the red mud as precipitation and do not enter the solution. Si, Fe, and Zn impurities can be effectively removed by dissolving industrial aluminum hydroxide with sodium hydroxide. Dilute hydrochloric acid was dropped into the reaction system for direct aging, and the solution balance was destroyed. The *γ*-AlO(OH) crystals were rapidly generated with fine crystal particles, leading to a feeble intermolecular force; as a result, the adsorption of impurities is weak. Finally, added an excess of ammonium salt solution, in which the dissociated NH_4_^+^ can replace sodium ions in the reaction process to generate Na_2_CO_3_ [21]. NH_4_^+^ can be absorbed by *γ*-AlO(OH), playing a role in deep denitrification. Ammonium ions on the surface of *γ*-AlO(OH) were released in the form of ammonia molecules during calcination to increase the specific surface area, which is conducive to the physical adsorption of carbon dioxide, and the remaining ammonium ions were grafted directly onto the alumina surface to facilitate the chemisorption of carbon dioxide. The main chemical reactions in the preparation process are as follows:Al(OH)_3_ + NaOH → NaAl(OH)_4_;(1)
SiO_2_ + 2NaOH → Na_2_SiO_3_ + H_2_O;(2)
2NaAl(OH)_4_ + 1.7Na_2_SiO_3_ → Na_2_O·Al_2_O_3_·1.7SiO_2_·*n*H_2_O ↓ + 3.4NaOH + 3H_2_O;(3)
(4)$$\mathrm{NaAl}(\mathrm{OH}{)}_{4}+\mathrm{HCl}\;\to \;{\mathrm{Al}(\mathrm{OH})}_{4}^{-}+\mathrm{NaCl}+H^{+};$$
(5)$${\mathrm{NaAlO}}_{2}\;\to \;\mathrm{NaOH}+{\mathrm{Al}(\mathrm{OH})}_{4}^{-};$$
NaOH + (NH_4_)_2_CO_3_ → Na_2_CO_3_ + NH_3_∙H_2_O;(6)
(NH_4_)_2_CO_3_ → 2NH_4_^+^ + CO_3_^2−^(7)
(8)$${\mathrm{Al}(\mathrm{OH})}_{4}^{-}\;\to \;\gamma -\mathrm{AlO}(\mathrm{OH})\cdot nH_{2}O+(1-n)H_{2}O;$$
2*γ*-AlO(OH)·*n*H_2_O → *γ*-Al_2_O_3_ + 2(1 − *n*) H_2_O.(9)

As shown in the above equations, a certain amount of hydrochloric acid was added to the sodium aluminate solution, and the mixed solution can be adjusted to the required pH values. It can generate the corresponding amount of NaCl. The amount of NaCl is directly proportional to the amount of the added hydrochloric acid while inversely proportional to the initial pH of the solution. In the context of industrial production, it is required to electrolyze saturated NaCl solution to prepare NaOH because the sodium ions in the mother liquor should be recycled to the circulating mother liquor, and high energy consumption in the electrolysis, complex process, and low efficiency are not conducive to its practical application. In this paper, the method of adding ammonium carbonate can replace the sodium in the solution with sodium carbonate. The conversion of sodium carbonate to sodium hydroxide needs the aid of limestone. Therefore, the sodium ions can be returned to the circulating mother liquor under the conditions of low cost and simple process. To summarize, the initial pH value of the solution should be controlled at a higher level to ensure that more sodium is returned to the circulating mother liquor.

It is shown in Table 8 that mesoporous *γ*-Al_2_O_3_ materials with high purity were synthesized by coprecipitation and direct aging method of sodium aluminate with hydrochloric acid. The impurity content, including SiO_2_, Fe_2_O_3_, Na_2_O, ZnO, of MA is lower than 0.01%; the removal efficiencies of SiO_2_, Fe_2_O_3_, Na_2_O, and ZnO are 99.746%, 99.805%, 95.833%, and 99.515%, and its total removal rate of impurities is 99.299%. Both the high specific surface area of 377.8 m^2^/g and the large pore volume of 0.55 cm^3^/g are beneficial to its catalytic activity and adsorption performance. Figure 5B–G show element mapping images of Al, Na, Fe, Si, and Zn. The distribution of impurities is extremely small, which is consistent with the small content of impurities in Table 8.

The XRD patterns of mesoporous *γ*-Al_2_O_3_ particles synthesized in a solution with pH 10 and calcined at 500 °C are shown in Figure 6A. It shows that main diffraction peaks (111), (220), (311), (222), (400), (511), and (440) of *γ*-Al_2_O_3_ with a cubic structure [49] (JCPDS Card No. 10-0425). All diffraction peaks present a high degree of broadness because of a fine nature and less degeneracy in the crystallites. Based on the Debye–Scherrer equation [51], *D* = *k λ*/*β* cos*θ*, where *k* is the constant, *θ* is the diffraction angle, *λ* is the X-ray wavelength, *β* is the full width at half-maximum (FWHM), and the principal crystallite size of *γ*-Al_2_O_3_ calculated from the full width at half-maximum of the isolated (311), (400), and (440) diffraction peaks are 13.8 nm, 12.9 nm, and 18.5 nm, respectively. As reported in the literature [52], solid metal oxides wielded for carbon capture have a large reduction in surface area after high-temperature treatment. In addition, only the thermodynamically stable (100) face with low coordinate corner/edge sites shows favorable binding to CO_2_, providing an inherently low capacity. The (111) facet, however, exhibits a high concentration of low coordinate sites. It was revealed that there are several intrinsic differences in the effects of sintering on basic site retention. In the presence of (111) facet, high-temperature treatment favors the retention of alkaline sites on the material surface.

According to the IUPAC classification [50], the nitrogen adsorption–desorption isotherms of powder material shown in Figure 6B are IV isotherms, which means that the fine powder is MA material with pore size ranging from 2 nm to 16 nm. For solid adsorbent materials, the desorption temperature is related to the alkaline strength, that is, the higher the desorption temperature is, the stronger the alkalinity is. The desorbed area indicates the number of alkaline sites, and the larger the desorbed area is, the larger the number of basic sites is. As shown in Figure 6F, after the adsorption of CO_2_ by MA powder, CO_2_ desorption peaks of different intensities were generated. Weakly basic desorption peaks were observed at 173.6 °C, medium basic desorption peaks at 394.4 °C, strongly basic desorption peaks at 537.6 °C, 565.0 °C, and 671.2 °C, respectively, which has the advantage of improving the CO_2_ adsorption performance of MA. As shown in Figure 6B XRD pattern, Figure 6F CO_2_-TPD profile, and Table 8, after the MA material calcinated at 500 °C for 4 h, there are more basic sites on its surface, and the specific surface area of the material is also large. The two factors together promote the capture of CO_2_. SEM morphology and size of the nanosized mesoporous *γ*-Al_2_O_3_ are shown in Figure 6G, which shows that size of the MA particles calcined at 500 °C are about 30–100 nm. Additionally, Figure 6H demonstrates the TEM image of as-synthesized MA material; it can be seen that there are plenty of pore structures on the surface of the fine powder, indicating that the material has a large specific surface area.

#### Characterization of CO_2_ Adsorption Properties

Up to now, how to tune volumetric CO_2_ capture performance remains unclear, and volumetric adsorption performance of materials is very important for their industrial application because it can determine the size and efficiency of the adsorption bed accommodating the absorbent. The volumetric adsorption capacity is the product of gravimetric adsorption capacity and density; therefore, the density has a large effect on volumetric adsorption performance [53]. Thus far, different densities for CO_2_ volumetric adsorption have been reported [54,55,56,57,58,59,60,61], such as true density, tap density, apparent density, and packing density. There are different results when studied at different densities, therefore, density was not chosen to be more convincing for the characterization of the adsorption properties of CO_2_. As reported in the literature [62], the dynamic CO_2_ adsorption performance of MA powder was investigated in a thermogravimetric analyzer (TGA Q500, TA Corp, New Castle, DE, USA). As shown in Figure 7A,B, comparing the CO_2_ adsorption properties measured by the presented CO_2_ adsorption measurement technique and gravimetric adsorption, it was found that the adsorption amounts determined by these two methods were essentially the same, indicating that the presented adsorption measurement can be employed.

Figure 7A shows the CO_2_ adsorption curve of MA material synthesized under optimal conditions. The adsorption capacity gradually increased for 24 min and then remained stable after 24 min. According to the pore-size distribution curve and results reported in the literature [45,63], the as-prepared MA powder is a mesoporous material with a pore size ranging from 2 nm to 16 nm, which is sufficient to accommodate CO_2_ molecules with kinetic diameters of 0.33 nm. After the ventilation with CO_2_, the gas molecules diffuse in the pore and contact the amino group for physical and chemical adsorption. With the extension of absorption time, more CO_2_ molecules will enter the pores of the MA material’s surface through physical adsorption. The amino group on the material surface will also interact with more CO_2_, making the adsorption amount increase. When the adsorption time is more than 24 min, most of the available amino reactions are completed, and the pore volume on the surface of the material is also explicit. The adsorption capacity gradually becomes stable. The adsorption properties of adsorbents prepared in this work were compared with those of the metal oxide and organic amine-modified materials, as shown in Table 9; it can be revealed that the MA adsorbent prepared in this study has a large adsorption capacity at low temperature and low flow rate. It is therefore a promising CO_2_ adsorbent. This is because the method contributes to the synthesis of MA with high purity and large specific surface area and enhances the alkaline sites on the MA material surface, thus improving the physical and chemical adsorption capacity of CO_2_.

Temperature is a pivotal parameter that has an important influence on the CO_2_ adsorption performance of surface-ammonium mesoporous material. The CO_2_ adsorption property of alumina at various temperatures under the gas flow rate of 20 cm^3^/min was investigated. As shown in Figure 7C, with the increase in temperature, from 30 °C to 150 °C, the adsorption capacity of CO_2_ decreases from 1.58 mmol/g to 0.85 mmol/g, but its change range is not considerable. According to the results in Figure 7C, and as reported in the literature [66], it is speculated that both physical adsorption and chemical adsorption occur on the surface of the material. When the temperature is low, physical adsorption predominates, and the physical adsorption process is exothermic; therefore, the adsorption amount decreases with the increase in adsorption temperature. When the temperature is high, chemical adsorption is mainly used. The physical adsorption process is exothermic, and therefore, the adsorption amount decreases with the increase in adsorption temperature. The reaction of amino with CO_2_ to generate carbonate is a reversible and exothermic reaction. With the rise in temperature, the reaction moves in the opposite direction, which is not conducive to the reaction of amino with CO_2_. However, CO_2_ is more likely to diffuse into the pore and react with amino and the chemical adsorption amount will gradually increase. Therefore, with the increase in temperature, the adsorption amount of CO_2_ decreases, but the change range of the adsorption amount of CO_2_ is imperceptible.

Figure 7D shows that the CO_2_ adsorption capacity of the same MA powder varies with different gas flow rates. As the gas flow rate increases, the CO_2_ adsorption capacity of MA first increases and then decreases. When the gas flow rate is small (10–20 cm^3^/min), the concentration of CO_2_ distributed around the alkaline sites on the MA surface dominates the adsorption process. As the gas flow rate increases, the CO_2_ concentration gradually increases, and the adsorption equilibrium shifts to the positive reaction direction; therefore, the CO_2_ adsorption capacity gradually increases. When the gas flow rate surpasses 20 cm^3^/min, the contact time between CO_2_ and the alkaline sites on the adsorbent surface plays a leading role in the adsorption process. As the gas flow rate continues to increase, the contact time between CO_2_ and the basic sites on the MA surface gradually decreases. Some CO_2_ molecules penetrate before interacting with the N atoms on the MA surface. CO_2_ adsorption cannot be completed successfully, and the amount of CO_2_ adsorption gradually decreases. Therefore, as the gas flow rate continues to increase, the amount of CO_2_ adsorption first increases from 1.01 mmol/g to 1.58 mmol/g and then decreases to 1.51 mmol/g.

In addition to the above, whether the CO_2_ adsorption capacity can be kept stable in the cyclic adsorption is a crucial index to measure the CO_2_ adsorption performance of the adsorbent, indicating that the structure of alumina is relatively stable and has an excellent cyclic performance. Figure 7E is a histogram of the CO_2_ adsorption capacity of MA material with the number of cycles. After eight cycles, the adsorption capacity and sample mass basically do not change with the increase in the number of cycles. However, during the second to eighth cycles of adsorption, the sample adsorption amount changed slightly, which may be due to incomplete desorption of CO_2_.

## 4. Conclusions

In this paper, hydrochloric acid was added to the sodium aluminate solution for pH value adjustment (pH value of 10). The precursor of *γ*-alumina material, *γ*-AlO(OH), was prepared by coprecipitation and direct aging methods. Ammonium ion was used in ammonium carbonate to remove residual sodium impurities in the precursor. After calcination at 500 °C for 4 h, the impurity of the MA material produced is less than 0.01%, with a large specific surface area of 377.8 m^2^/g. The pore volume is 0.55 cm^3^/g, and the average pore size is 3.1 nm. These are favorable parameters of porous alumina application in adsorbents, filters, catalysts, and catalyst supports. The CO_2_ adsorption capacity is 1.58 mmol/g at a gas flow rate of 20 cm^3^/min.

## Figures and Tables

**Figure 1 materials-14-05465-f001:**
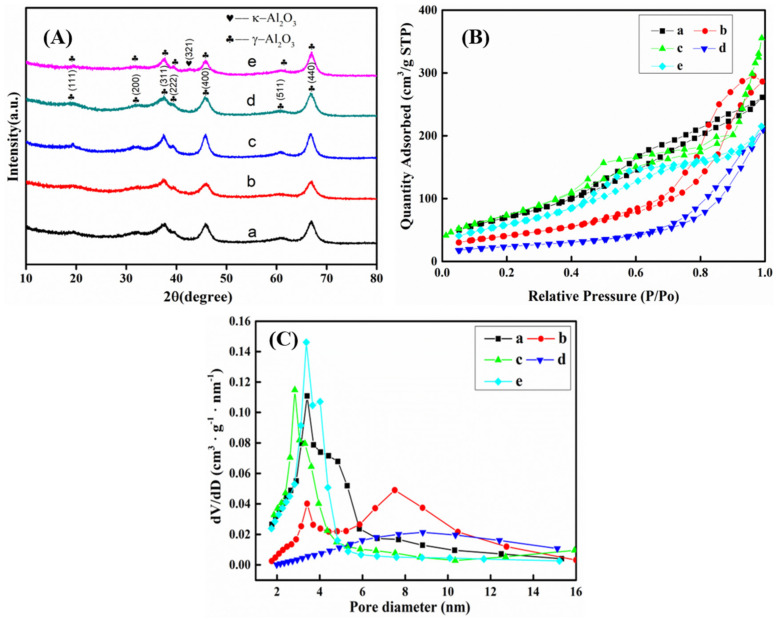
XRD pattern (**A**), nitrogen adsorption–desorption isotherms (**B**), and pore-size distribution curves (**C**) of the MA synthesized in various pH solutions and calcined at 500 °C: (a) pH = 6; (b) pH = 8; (c) pH = 10; (d) pH = 12; (e) pH = 13.

**Figure 2 materials-14-05465-f002:**
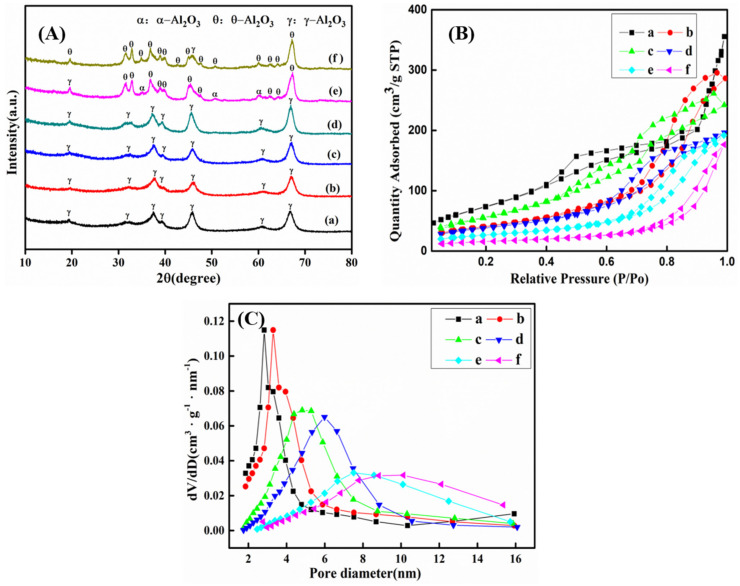
XRD pattern (**A**), nitrogen adsorption-desorption isotherms (**B**), and pore-size distribution curves (**C**) of the precipitate heat treated at various temperatures: (a) 500 °C; (b) 600 °C; (c) 700 °C; (d) 800 °C; (e) 900 °C; (f) 1000 °C.

**Figure 3 materials-14-05465-f003:**
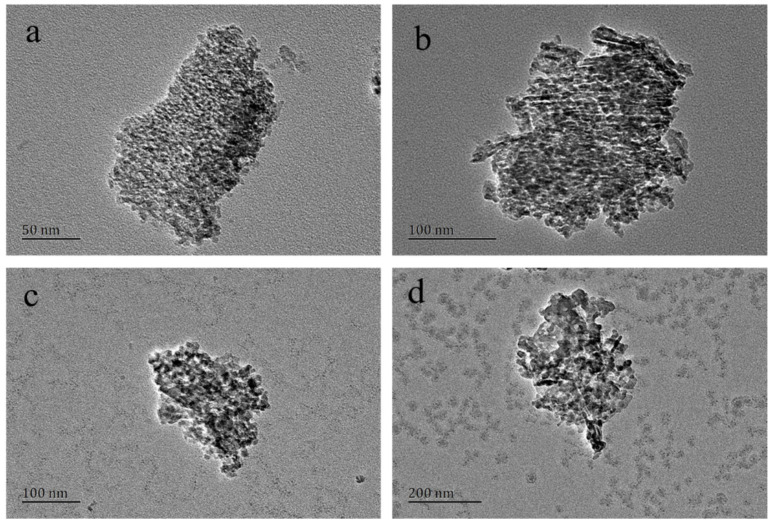
TEM images of the MA powders calcined under various temperatures: (**a**) 500 °C; (**b**) 700 °C; (**c**) 900 °C; (**d**) 1000 °C.

**Figure 4 materials-14-05465-f004:**
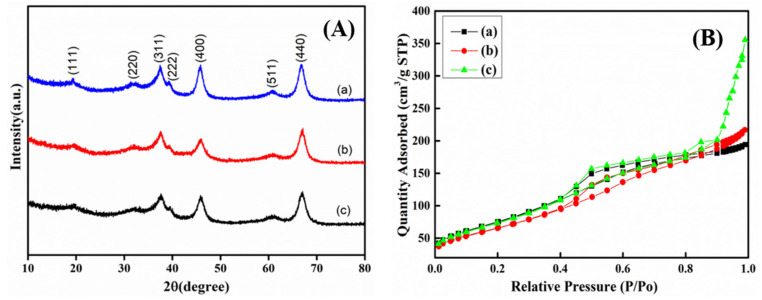
XRD pattern (**A**) and nitrogen adsorption–desorption isotherms (**B**) of products synthesized with different desodium agents.

**Figure 5 materials-14-05465-f005:**
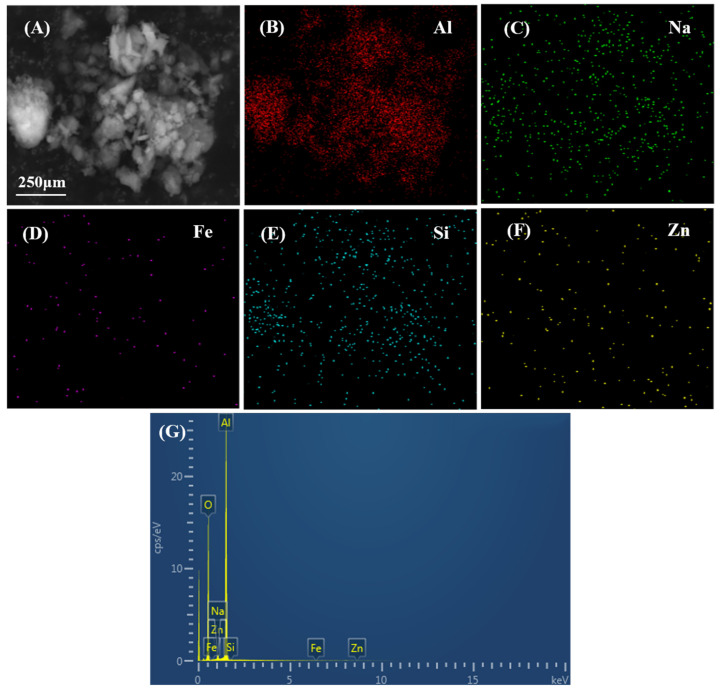
(**A**) SEM image; (**B**–**F**) element mapping images of Al, Na, Fe, Si, and Zn; (**G**) EDS spectra of element mapping images.

**Figure 6 materials-14-05465-f006:**
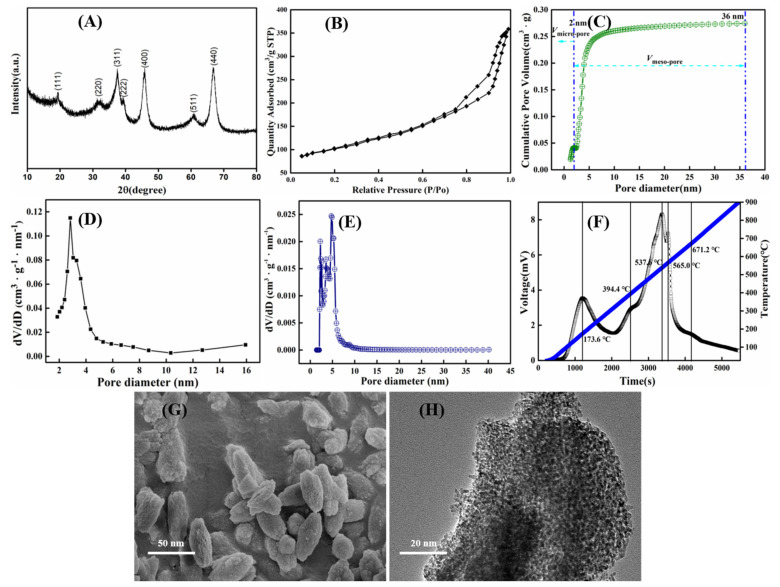
(**A**) XRD pattern; (**B**) nitrogen adsorption–desorption isotherm and pore-size distribution curve; (**C**) determination of pore volume obtained by NLDFT; (**D**,**E**) pore-size distribution curves through BJH model and NLDFT kernel; (**F**) CO_2_-TPD profile; (**G**) SEM image; (**H**) TEM image of *γ*-Al_2_O_3_ materials synthesized at pH = 10 by calcining precipitation of sodium aluminate and hydrochloric acid at 500 °C.

**Figure 7 materials-14-05465-f007:**
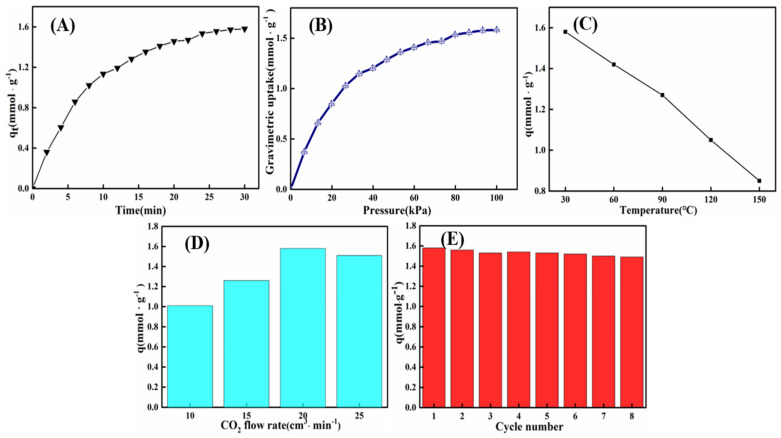
(**A**) Curve of CO_2_ adsorption via presented measure; (**B**) curve of gravimetric CO_2_ adsorption; (**C**) graph of adsorption capacity at different temperatures; (**D**) column chart of adsorption capacity of materials at different gas flow rates; (**E**) curve of CO_2_ adsorption capacity in different cycles of *γ*-Al_2_O_3_ material synthesized at pH = 10 by calcining precipitation of sodium aluminate and hydrochloric acid at 500 °C.

**Table 1 materials-14-05465-t001:** Components of industrial aluminum hydroxide.

Component Content (*w*_B_%)				
SiO_2_	Fe_2_O_3_	Na_2_O	ZnO	Al(OH)_3_
0.67	1.59	0.36	0.66	96.72

**Table 2 materials-14-05465-t002:** Effect of solution pH value on physical properties of the nanosized powder.

pH Value of Solution	Specific Surface Area (m^2^/g)	BJH Adsorption Summary	
		Pore Volume (cm^3^/g)	Pore Size (nm)
6	263.4	0.34	4.4
8	348.3	0.29	6.6
10	377.8	0.55	3.1
12	283.7	0.18	7.6
13	221.8	0.25	3.9

**Table 3 materials-14-05465-t003:** CO_2_ adsorption–desorption performance of industrial alumina and MA prepared in different pH solutions: (a) pH = 6; (b) pH = 8; (c) pH = 10; (d) pH = 12; (e) pH = 13.

Samples	Gas Flow Rate	Adsorption Capacity	Desorption Capacity
	(cm^3^/min)	(mmol/g)	
Industrial alumina	20	0.048	0.047
(a)	20	0.68	0.67
(b)	20	1.13	1.12
(c)	20	1.58	1.56
(d)	20	0.84	0.82
(e)	20	0.66	0.65

**Table 4 materials-14-05465-t004:** Effect of calcination temperatures on physical properties of the nanosized powder.

Calcination Temperature (°C)	Specific Surface Area (m^2^/g)	BJH Adsorption Summary	
		Pore Volume (cm^3^/g)	Pore Size (nm)
500	377.8	0.55	3.1
600	316.5	0.39	4.3
700	263.6	0.30	5.6
800	188.7	0.26	6.2
900	146.5	0.23	8.2
1000	94.8	0.11	10.3

**Table 5 materials-14-05465-t005:** CO_2_ adsorption–desorption performance of industrial alumina and MA calcined in various temperatures: (a) 500 °C; (b) 600 °C; (c) 700 °C; (d) 800 °C; (e) 900 °C; (f) 1000 °C.

Samples	Gas Flow Rate	Adsorption Capacity	Desorption Capacity
	(cm^3^/min)	(mmol/g)	
Industrial alumina	20	0.048	0.047
(a)	20	1.58	1.56
(b)	20	1.26	1.24
(c)	20	0.84	0.83
(d)	20	0.66	0.66
(e)	20	0.22	0.21
(f)	20	0.067	0.065

**Table 6 materials-14-05465-t006:** Effect of desodium agent on physical properties of the nanosized powder.

Desodium Agent	Specific Surface Area (m^2^/g)	BJH Adsorption Summary	
		Pore Volume (cm^3^/g)	Pore Size (nm)
Agent Ⅰ	278.1	0.44	3.06
Agent Ⅱ	248.3	0.47	3.24
(NH_4_)_2_CO_3_	377.8	0.55	3.06

**Table 7 materials-14-05465-t007:** CO_2_ adsorption–desorption performance of industrial alumina and MA prepared via different desodium agents: (a) NH_4_Cl; (b) CH_3_COONH_4_; (c) (NH_4_)_2_CO_3_.

Samples	Gas Flow Rate	Adsorption Capacity	Desorption Capacity
	(cm^3^/min)	(mmol/g)	
Industrial alumina	20	0.048	0.047
(a)	20	0.78	0.77
(b)	20	1.03	1.03
(c)	20	1.58	1.56

**Table 8 materials-14-05465-t008:** Content of impurity in *γ*-Al_2_O_3_ powder and its physical properties.

Impurity	SiO_2_	Fe_2_O_3_	Na_2_O	ZnO	Total Values	Specific Surface Area (m^2^/g)	Pore Volume (cm^3^/g)	Pore Diameter (nm)
Content (*w*_B_%)	0.0017	0.0031	0.015	0.0032	0.023	377.8	0.55	3.1
Removal efficiency (%)	99.746	99.805	95.833	99.515	99.299			

**Table 9 materials-14-05465-t009:** CO_2_ adsorption performance of various adsorbents.

Adsorbents	Temperature (°C)	Gas Flow Rate (cm^3^/min)	CO_2_ Uptake (mmol/g)	Reference
Industrial alumina	30	20	0.048	Present study
MA	30	20	1.58	Present study
TiO2-TEPA	30	10	1.63	[64]
PEI/Zr7-SBA-15	75	20	1.56	[65]
PAA-alumina	110	90	1.07	[38]
PEI-alumina	110	90	1.87	[38]
MA-Bh	25	-	0.58	[40]
MA-Bh	120	-	2.17	[40]
Bh-AO13	25	-	0.45	[40]
Bh-AO13	120	-	3.84	[40]

## Data Availability

The data used to support the findings of this study are available from the corresponding author upon request.

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
