# Peer review of "Study on the Synthesis of High-Purity γ-Phase Mesoporous Alumina with Excellent CO2 Adsorption Performance via a Simple Method Using Industrial Aluminum Oxide as Raw Material"

_materials, 2021, doi:10.3390/ma14195465_

Round 1

Reviewer 1 Report

The authors have addressed all the comments raised previously, I now recommend the publication of this work.

Author Response

Thanks for your advice, please see the attachment.

Reviewer 2 Report

The manuscript reports the synthesis of high purity γ-phase mesoporous alumina and the study of its CO2 adsorption performance.

The research has been performed well and a complete morphological and structural characterization of the materials has been presented. Furthermore, the CO2 adsorption ability of the studied materials, even in different cycles, has been appropriately reported.

The latest version of the manuscript is certainly better than the previous one and its scientific quality has been now raised to a level appropriate for publication in Material journal.

The paper is well written, the text is clear and easy to read, and the introduction is sufficient.  However, the manuscript could be accepted for publication after minor revision and the following issues should be taken to improve its quality:

- The MA acronym was used in the title, but its meaning is clarified in the Abstract. In the title, Authors should avoid using abbreviations and non-standard symbols.

- Please correct the typo in line 31 (methodmethod).

- The originality of the research is adequate, however the novelty with respect to the considerations presented in literature should be emphasized.

- The Table 1 in line 325 (Content of impurity in γ-Al2O3 powder and its physical properties) and the Table 2 in line 388 (CO2 adsorption performance of various adsorbents) should be renominate with a different numbering (Table 8 and Table 9) in order not to confuse them with the previous Tables 1 and 2.

- The sentence in lines 418-419 seems to be incomplete. I consider that it is advisable to better explain why the MA adsorbent prepared in this study is better than those reported in the literature.

Author Response

(The authors gave the same response as above.)

Reviewer 3 Report

The authors study CO2 adsorption by γ-phase mesoporous alumina synthesized from industrial aluminum oxide. There were some problems in this manuscript as follows.

  1. Unfortunately the manuscript is poorly written and difficult to understand. Extensive editing of the paper is required for clarity: section Introduction, lines 106-107, 154-156, 176-178, 219-222, 270-286, 321-325, 355-361 and others.
  2. Line 54: “greatly increasing the industrial alumina adsorption capacity (from 0.7 to 0.07 mmol/g” - Please check
  3. Line 60: 2. What is “MA-Bh”?  Please expand it.
  4. Lines 162-163: “When 161 the pH value exceeds 10, the specific surface area decreases from 377.8 m2 /g to 221.8 m2 /g 162 and the pore volume decreases from 0.55 cm3 /g to 0.25 cm3 /g” – Please, explain this result.
  5. Lines 174-175: “The large specific surface area per unit volume can give more basic sites on the surface of the MA material” – The sentence is unclear.
  6. It seems more convenient to use continuous numbering of samples throughout the manuscript.
  7. 3: Can't we have a more magnification picture in order to appreciate the actual structure of the synthesized particles?
  8. Table 6: What is “agent”?
  9. Lines 255-265: 6. The discussion of the effect of different desodium agents to the CO2 adsorption capacity needs improvement.
  10. Chemical reactions (2), (5), (10) are in doubt. The authors need to provide some literature or supporting evidence if available.
  11. Please, check the significant figures in the numbers along all the research and use a confidence interval for the mean instead of a single number for the mean (tables, figures)
  12. Сheck the numbering of tables and links to them through all manuscript.

Author Response

We all thank you for your comments concerning our manuscript, please see the attachment

This manuscript is a resubmission of an earlier submission. The following is a list of the peer review reports and author responses from that submission.

Round 1

Reviewer 1 Report

The manuscript is generally sound, but I have two key concerns.

First, the authors discuss a lot of XRD data but do not make specific links to the adsorption behaviour. It is known from other related systems (e.g. MgO) that the specific exposed facet of the adsorbent plays a major role in adsorption behaviour and its relationship with sintering/changes to the surface. For example, see https://pubs.acs.org/doi/abs/10.1021/jacs.8b01845. There are often intricate relationships between e.g. facet, surface area, pore size and sorption capacity which are poorly explained here.

Secondly, and related to the first point, there are no surface sensitive techniques employed to probe the specific nature of the interaction of carbon dioxide with the surface. Again, is it specific facets that give rise to the sorption, or is it specific sites? Generally, the detail on this aspect is lacking.

Also, the description of the loss of capacity vs cycling is incomplete. It should not be described as a manual error - the source of the decrease should be understood and interpreted within the framework of existing processes, e.g. calcium looping.

Finally, it is not clear how the capacity of this material compares to other materials (metal oxides, amine supported solids etc).

Otherwise the manuscript presents an understandable synthesis procedure and appropriate characterisation.

Author Response

Thanks for your revision comments, Please see the attachment to get our reply to the review report.

Reviewer 2 Report

This work reported the synthesis of a high purity γ-phase mesoporous alumina (MA) with large specific surface area, porosity and excellent CO2 adsorption performance. The conditions of synthesis, such as PH value, aging time, calcination temperature and desodium agent were investigated. I recommend the publication of this work if the following comments can be addressed:

  1. Why did the high PH value (PH=13) lead to the formation of k-Al2O3?
  2. The TEM images in Figure 4 are not conclusive enough to show the particle size of the mesoporous alumina. High resolution images with particle size distributions measured from at least 100 particles are recommended to study the effect of calcination temperature on the particle size.
  3. Please add the types of desodium agents in the caption of Figure 4
  4. EDS spectra of Figure 5 should be provided with the elemental mapping. ICP analysis is recommended to study the impurity of the alumina.
  5. There are some typos in the manuscript, such as Page 7 line 191, Fig. 4 should be Fig. 3.

Author Response

Thanks for your revision comments, please see the attachment to get our reply to the review report.

Reviewer 3 Report

This article addresses the synthesis of mesoporous alumina and assessment of their textural and CO2 adsorption characteristics.

The following major revisions should be done:

-clear explanation of the methodology of CO2 adsorption measurement and comparison of the results with more conventional adsorption isotherm measurement techniques (at least relying on the literature)

-equilibrium pressure at which adsorbed amounts were measured should be provided

- BJH model is no longer considered for mesopore size analysis, it would be better to use appropriate NLDFT kernel

Article would benefit from extensive editing of English, to avoid phrases like in page 4, lines 138-140: « A subsequent long adsorption platform indicates that the adsorption of N2 in the capillary is saturated. P wind is greater than O. The adsorption after 9 is caused by the agglomeration of N2 between large particles.”

Author Response

Thanks a lot for your revision comments. Please see the attachment to get ur reply to the review report.

Round 2

Reviewer 1 Report

.

Reviewer 2 Report

Thank you for addressing my concerns thoroughly, I can now recommend the publication of the
manuscript.